# Recent Progress and Applications of Thermal Lens Spectrometry and Photothermal Beam Deflection Techniques in Environmental Sensing

**DOI:** 10.3390/s23010472

**Published:** 2023-01-02

**Authors:** Mladen Franko, Leja Goljat, Mingqiang Liu, Hanna Budasheva, Mojca Žorž Furlan, Dorota Korte

**Affiliations:** Laboratory for Environmental and Life Sciences, University of Nova Gorica, Vipavska 13, 5000 Nova Gorica, Slovenia

**Keywords:** thermal lens microscopy, beam deflection spectrometry, microfluidic system, microcystin-LR detection, iron species determination, ammonium detection

## Abstract

This paper presents recent development and applications of thermal lens microscopy (TLM) and beam deflection spectrometry (BDS) for the analysis of water samples and sea ice. Coupling of TLM detection to a microfluidic system for flow injection analysis (μFIA) enables the detection of microcystin-LR in waters with a four samples/min throughput (in triplicate injections) and provides an LOD of 0.08 µg/L which is 12-times lower than the MCL for microcystin-LR in water. μFIA-TLM was also applied for the determination of total Fe and Fe(II) in 3 µL samples of synthetic cloudwater. The LODs were found to be 100 nM for Fe(II) and 70 nM for total Fe. The application of µFIA-TLM for the determination of ammonium in water resulted in an LOD of 2.3 µM for injection of a 5 µL sample and TLM detection in a 100 µm deep microfluidic channel. For the determination of iron species in sea ice, the BDS was coupled to a diffusive gradient in the thin film technique (DGT). The 2D distribution of Fe(II) and total Fe on DGT gels provided by the BDS (LOD of 50 nM) reflected the distribution of Fe species in sea ice put in contact with DGT gels.

## 1. Introduction

In recent years, many different photothermal (PT) techniques have been developed, offering promising spectrometric detection methods that have found application in many fields of applied science, including environmental sensing (ES) [1,2,3]. The main goal of ES is to collect information about the anthropogenic impact on the environment and help reduce it to a minimum by monitoring the air, soil and water quality, which further helps understand and manage the risks to biota and human health. Thus, there is a continuous and expanding need for developing sensors that provide improvements regarding sensitivity as well as selectivity compared to techniques and methodologies based on traditional chemical analysis of environmental samples. Furthermore, the advanced sensors should not require complicated sample preparation procedures and shall be easy to use and provide high sample throughput with fast readout. Considering all these requirements, PT techniques seem to be a good basis for well-applicable analytical methods for a number of environmental sensing applications. The interest in PT techniques is increasing further by recent developments in laser technology, which resulted in compact diode or solid-state lasers [4] that are smaller in size but of higher output power and with a wider range of output wavelengths as compared to gas lasers, which on the other hand require a cooling system with large consumption of water. All this, in combination with low cost and flexibility of fiber optics-based laser systems, makes PT sensors portable, versatile, and easy to use is already well established, but also in new emerging fields of environmental sensing and monitoring [5,6,7,8,9]. It is thus quite expected that the use of PT techniques in ES continues to grow.

In this work, the focus is on recent progress and applications of thermal lens spectrometry (TLS) and beam deflection spectrometry (BDS), which are best suited for the analysis of condensed samples among all PT techniques since they are based on probing the changes of refractive index with temperature, which are induced by the absorption of light in the sample. The temperature coefficient of the refractive index of liquids is much higher than for gasses; therefore, the sensitivity of detection in liquid samples is higher by TLS and BDS, compared to photoacoustic spectroscopy (PAS), which exploits the pressure change in the irradiated sample [10]. PAS is clearly the PT technique of choice for gas sensing, while some applications of PAS for sensing higher pollutant concentrations in water were reported as well [11,12]. Recently, TLS and BDS were successfully applied for environmental monitoring and investigations of environmental processes and conditions by determining specific compounds and ions in natural and wastewaters and for speciation studies of elements in waters and sediments. The range of analytes determined by TLS in environmental samples ranges from toxic compounds and elements, such as for example pesticides, colloidal Ag and Cr(VI), or markers/indicators of undesired environmental processes like algal blooms (carotenoids, phthalocyanines), degradation of organic matter (ammonia, biogenic amines) or anoxic conditions in aquatic systems (Fe(II)/Fe(III)) [13,14,15,16,17,18,19]. Several other applications of TLS in the analysis of environmental and related samples can be found in previous review papers and book chapters [20,21,22,23,24,25]. On the other hand, the number of applications of BDS for analysis and characterization of environmental samples is quite limited and, so far, related mainly to the characterization of diffusive gradient thin film (DGT) based passive samplers and speciation of iron in sediments [26] and to the determination of H_2_S by exploiting beam deflection in response to temperature variations above plasmonic nanostructures resulting from adsorption of the analyte on Au nanoparticles [27]. A microcantilever sensor based on BDS, which provides a miniature sensing platform for the real-time, simultaneous detection of multiple target analytes using a single device, was also proposed and offered potential applications in environmental sensing as well [28].

Both TLS and BDS are based on photoinduced changes in the thermal state of the sample. Light energy absorbed by the sample is released in the form of heat due to non-radiative relaxation processes, which induce temperature changes in the sample and its immediate surroundings. TLS is based on probing the changes in samples’ refractive index, which are introduced by the temperature rise in the illuminated sample, and cause defocusing of the probe beam, whereas BDS relies on probing the temperature oscillations (TOs) induced in the fluid close to the sample surface, through the periodic deflections of the probe beam, which follow the oscillations of the refractive index gradient in the fluid as a result of the TOs.

Basic equations, which relate the measured PT signal to the concentration of the analyte and to the experimental parameters, are given in the continuation. Thermal lens signal is usually expressed as the relative change in the axial intensity of the probe beam (ΔI/I), which is directly proportional to the concentration of the analyte (C), the power of the excitation beam (P), and to the temperature coefficient of the refractive index (dn/dT) of the sample, as given in Equation (1):(1)$$\frac{\Delta I}{I}=-\frac{\mathrm{dn}}{\mathrm{dT}}\frac{2.303P\varepsilon \mathrm{Cb}}{\lambda k}$$
where λ is the probe beam wavelength, k is the thermal conductivity of the sample, ε is the molar absorptivity of the analyte (expressed in L/(mol·cm)) at the wavelength of the excitation laser, and b is the optical interaction length (i.e., sample thickness). It should be noted that Equation (1) is valid for the steady state thermal lens conditions, meaning that the duration of the excitation is much longer than the characteristic time of the thermal lens, and for the experimental configuration where the sample is located √3 confocal distances behind the focal point of the probe beam, and the thermal lens is generated by a pump beam with a radius equal to the probe beam radius in the sample.

In the case of the BDS technique, the signal is proportional to the deflection angle (φ) of the probe beam with respect to the initial propagation (without TO), which is given by Equation (2).
(2)$$\varphi =-\frac{1}{n_{0}}\frac{\partial n}{\partial T}\frac{\beta P}{\sqrt{\pi }k}\frac{1}{\sqrt{e}}$$

Here n_0_ is the refractive index of the fluid, β is the absorption coefficient of the sample (expressed in cm^−1^), which is related to the analyte concentration, and e is the base of the natural logarithm. Other parameters were defined already for Equation (1). Moreover, Equation (2) is valid only for the steady state conditions and for an experimental configuration where the axes of the probe and the pump beam do not cross but are offset by a/2, where a is the radius of the excitation beam at the sample surface.

Since the signal in the case of TLS and BDS is directly proportional to the absorbed optical energy and, therefore, proportional to the concentration of the analyte and to the power of the excitation laser, these techniques enable ultra-sensitive chemical analysis (corresponding to the detection of absorbances on the order of 10^−7^ absorbance units). Furthermore, tight focusing of laser beams (close to diffraction limit) enables probing of sample volumes on the order of a few fL and characterization of small surfaces (about 1 μm^2^) of solid samples. Finally, the characteristic signal response time (time constant) in liquids is on the order of some 10 milliseconds, which makes it possible to apply such techniques as detectors in flow systems, like those encountered in liquid chromatography, flow injection analysis (FIA), and microfluidics [13,29,30,31,32].

It is the purpose of this paper to illustrate how the above-mentioned capabilities and characteristics of PT techniques can be exploited for highly selective and sensitive chemical analysis of low-volume environmental samples, with a particular focus on fast screening systems and determination of chemically unstable analytes, which do not allow for sample treatment prior the analysis.

## 2. Instrumental

Basic instrumental schemes and configurations for TLS and BDS spectrometers have been described previously in original articles as well as in several reviews and book chapters [23,33,34,35,36,37].

In this chapter, we will only give a description of general TLM and BDS schemes used in applications presented in this paper. Slight modifications were made on TLM for the determination of particular analytes, including microcystin-LR, Fe(II) or total Fe, and ammonium, and will be outlined when discussing these applications.

### 2.1. Thermal Lens Microscope and Microfluidic System

Instrumentation for thermal lens microscopy (TLM), which from an instrumental point of view differs from pump/probe TLS spectrometers primarily by the fact that a single microscopic objective lens is used to focus the pump as well as the probe laser beams, has been reviewed previously [34,36,37].

The TLS microscope (TLM) used in this work is schematically depicted in Figure 1a. The excitation beam (EB) is provided by a DPSSL laser (LSR-PS-SA, Shenzhen 91 Laser Co., Ltd., Shenzhen, China) with an 80 mW output power at 532 nm. It is directed by a set of mirrors (reflection 99.8% at 45° from 400 to 580 nm, Thorlabs Inc., not shown in the figure), which facilitate the alignment of the pump and probe beams through a beam expander that is constructed of two convex lenses of focal lengths 30 and 50 mm (Bi–Convex, AR Coated: 350–700 nm, Edmund Optics, Barrington, NJ, USA) and modulated by a variable-speed mechanical chopper (Scientific Instruments, Control unit model 300C, chopping head model 300CD, chopping disks model 300H) at 1 kHz. The probe beam (PB) originating from a He-Ne laser (632.8 nm, 5 mW, 25-LHP-151–230, Melles Griot, Carlsbad, CA, USA) is first directed onto a beam expander (consisting of two convex lenses of focal lengths 40 and 150 mm, Bi-Conv ex, AR Coated: 350–700 nm, Edmund Optics), and combined with the EB by the use of a dichroic mirror (transmitting from 400 to 580 nm, reflecting from 620 to 780 nm, both at 45°, NT69-204, Edmund Optics). EP and PB propagate coaxially through a 15 mm focal length objective lens (Olympus Inc., Tokyo, Japan) and further through a Y-junction microfluidic chip (Part No. 3200008, The Dolomite Center, Royston, UK), where the sample is delivered, and a thermal lens is generated. The TLM signal is detected by an amplified photodiode detector (PDA-36A-EC, Thorlabs Inc., Newton, NJ, USA) equipped with a pinhole and an interference filter (CWL 633 nm, Melles Griot) and connected to a lock-in amplifier (Stanford research instruments, Model SR830 DSP) and PC for data acquisition.

The microfluidic (μFIA) system (b) is operated by a set of two microsyringe pumps (NE-1000, New Era Pump Systems Inc., Farmingdale, NY, USA), shown on the left-hand side of (a). One microsyringe pump is equipped with a 5 mL syringe that provides the carrier flow (5–20 μL/min). The second microsyringe pump is equipped with a 0.5 mL syringe and is used to deliver instantly (with a flow rate of 50–100 μL/min) a few μL of a sample or specific reagents into the Y-junction channel of the microreactor chip, which is further connected to the detection microchip positioned on the xyz translation stage.

### 2.2. Beam Deflection Spectrometer

The BDS measurements were performed using the experimental setup presented in Figure 2a.

A solid-state laser of 30 mW output power at 532 nm (BWI-532-10-E/66966) was used as a source of the EB. A He-Ne laser (Uniphase, Model 1103P, 632.8 nm, 3 mW) was used as a PB source. Each beam was focused by a 100 mm focal point lens (Bi-Convex, AR Coated: 350–700 nm, Edmund Optics). A variable-speed mechanical chopper (Scientific Instruments, Control unit model 300C, chopping head model 300CD, chopping disks set model 300H) was used to modulate the EB at a frequency of 3 Hz to ensure the collection of information from the bulk of investigated samples (thermal diffusion length equals the samples’ thickness) and to provide the maximum S/N ratio. The PB was directed twice through the area of TOs by a set of mirrors (400–750 nm, Thorlabs, model BB1-E02) to increase the length of interaction with the field of induced TOs. In turn, this provided higher sensitivity to the setup [38]. Deflection of the PB is detected by a quadrant photodiode (RBM–R. Braumann GmbH, Model C30846E) equipped with an interference filter (CWL 633 nm, Edmund Optics) and connected to a lock-in amplifier (Stanford research instruments, Model SR830 DSP), where the BDS signal and the signals’ phase are deconvoluted. The examined sample was placed on a 3D translation stage (CVI, Model 2480M/2488) to vary its position in *x*, *y,* and *z* direction and to optimize the experimental configuration.

Gels investigated in this work are sensitive to solvents and particularly incompatible with organic solvents, which would provide higher sensitivity because of large dn/dT. Air was, therefore, used as the contact fluid despite its low dn/dT value, which is, on the other hand, compensated in part by about 10 times lower k value, as compared to most organic liquids. The arrangement of the sample position with respect to the support and the fluid is schematically presented in Figure 2b.

## 3. Results and Discussion

### 3.1. Determination of Microcystin−LR in Water by µFIA-TLM

One of the most impressive demonstrations of the high sensitivity and high sample throughput capabilities of PT techniques in ES was the application of µFIA-TLM for the determination of cancerogenic Cr(VI) in waters [14], which provided a 20 samples/min throughput while keeping the LOD about 10 times lower as compared to spectrophotometry.

In this work, we have exploited a similar analytical platform for the determination of microcystin-LR, the most toxic neurotoxin among microcystins, arising from algal blooms of cyanobacteria [39], which are recently becoming more and more frequent as a consequence of pollution of aquatic ecosystems by nutrients as well as because of climatic changes.

Cyanotoxins, including microcystin-LR, are known inhibitors of phosphatase 2A (PP2A). The inhibition of PP2A is reflected in the reduced rate of the formation of p-nitrophenolate resulting from the p-nitrophenilphosphate (pNPP), a PP2A substrate.

p-nitrophenolate absorbs in the 400 nm range, which was exploited for the detection of the reaction product on the microchip, as shown in Figure 3.

The excitation beam from an Ar-ion laser operating at 458 nm wavelength (60 mW) was used in this case instead of the solid-state laser. The excitation beam was modulated at 3 kHz, which provided the best S/N ratio under given experimental conditions (excitation laser, carrier flow). The carrier solution consisted of a 3 mM pNPP and was delivered to the microreactor chip at 8 µL/min carrier flow rate. Samples were preincubated (15 min) with PP2A by adding 0.5 µL of PP2A solution (25.6 mU/mL) to 49.5 µL of the sample or standard microcystin-LR solution just prior to injection of 1 µL of preincubated sample into the carrier stream (reagent) with a 50 µL/min injection flow.

The formation of p-nitrophenolate by the PP2A enzymatic reaction was found to be eight times faster on the microchip compared to the microtiter plates used in ELISA PP2A assays. However, differently from the case of Cr(VI), where the TLM signal was increasing with the concentration of the analyte, a higher concentration of microcystin-LR means a lower activity of PP2A and consequently lower concentration of produced p-nitrophenolate, therefore lower TLM signal. This is depicted in Figure 4, which shows µFIA-TLM signals for six replicate injections of microcystin-LR standards at five different concentration levels (50 ng/L–1 µg/L), which served to prepare the calibration curve.

The presented results indicate good reproducibility, which is reflected in a 3.6% RSD, while the calibration curve showed a linear range from 0.08 to 1 µg/L. The calculated LOD (S/N = 3 basis) was 0.08 µg/L, which is 12 times lower than the maximum allowed microcystin-LR level in the water (1 µg/L) as recommended by the WHO. The performance characteristics of the presented µFIA-TLM PP2A assay for microcystin-LR compare favorably in terms of LOD with commercial microcystin kits and previously reported assays such as immunoelectrodes, as it is summarised in Table 1.

Test on spiked samples with microcystin-LR concentration of 300 ng/L revealed recoveries of 90 ± 14% for filtered samples and 93 ± 12% for nonfiltered samples. It should also be noted that the µFIA-TLM PP2A assay consumes over 100 times less PP2A enzyme per analyzed sample, while it produces less than 10 µL of liquid waste per analyzed sample (for triplicate injection), as compared to about 1 mL/sample in case of ELISA assays relying on microtiter plates [40,41]. µFIA-TLM PP2A cannot compete in terms of LOD with the immunoelectrode, which is, however, hindered by the linear range, requiring dilution of samples with microcystin-LR concentrations with concentrations higher than 0.1 μg/L (MCL = 1 μg/L), as well as by low sample throughput (<2 samples/h).

### 3.2. Determination of Total Fe and Fe(II) in Synthetic Cloudwater by µFIA-TLM

An interesting and challenging analytical problem, which requires low limits of detection and small volume capability, is the analysis of cloud water (the liquid phase of clouds), which is of significant importance for an understanding of cloud chemistry and particularly the associated photochemical processes, oxidation reactions, acidification of rain, and aerosol sulfate formation [43]. Average sample volumes of collected cloud water per sampling campaign are about 50 mL [44], which are further divided into subsamples for numerous physicochemical and chemical analyses (TOC, organic and inorganic ions, H_2_O_2_, microbiological analysis, etc.). Accordingly, volumes intended for each specific analysis should be reduced to a minimum. Total iron concentrations in cloudwater are reported to be between 0.1 and 9.1 µM [43]. Most sensitive analytical techniques for the determination of iron, such as ET−AAS or ICP−MS, can provide only information on total iron concentration. On the contrary, techniques suitable for Fe(II)/Fe(III) speciation studies (i.e., UV−Vis spectrometry or electroanalytical techniques) do not provide the required sensitivity for determination of Fe(II) and Fe(III) in small (few µL) volume samples. For example, with spectrophotometric measurements in a standard 1 cm × 1 cm cell, which requires at least a few mL of sample for one replicate, even determination of total Fe would not be possible at concentrations below 1 µM (molar absorptivity of ferroin = 11.000 M^−1^ cm^−1^). Therefore a method, which could offer selective determination of Fe(II) and Fe(III) in the sub-micromolar concentration range and would require lower, eventually below 10 µL sample volumes would significantly facilitate analysis and open new possibilities in further studies of cloudwater chemistry.

To elucidate the advantages and performance characteristics of µFIA-TLM and to validate the methods for Fe(II) and total Fe determination in µL size samples of cloudwater by µFIA-TLM, an analysis of spiked synthetic cloudwater was performed.

Synthetic cloudwater (marine type) was prepared according to Vaitilingom et al. [45] by dissolving 0.0128 g NH_4_NO_3_, 0.0040 g MgCl_2_ × 6H_2_O, 0.0234 g NaCl, 0.0018 g K_2_SO_4_, 0.0096 g CaCl_2_ × 2H_2_O, 5.800 mL of 0.005 M formic acid, 0.400 mL of acetic acid (99.8%), 0.300 mL 0.0100 M of succinic acid, 0.120 mL 0.05 M oxalic acid, 0.0088 g of NaOH, and 6.300 mL of 0.0100 M H_2_SO_4_ in 1.000 L of double-deionized H_2_O and then diluting up to 2.000 L. Spiked synthetic marine cloudwater was prepared by adding appropriate amounts of Fe(II) and Fe(III) stock solutions to marine cloudwater.

The carrier stream consisted of 30 mM 1,10-phenanthroline (*o*–phen) in deionized water and was operated at a 5–20 µL/min flow rate. 3 µL sample injection volume (at 100 µL/min injection flow rate) was found optimal. For the determination of Fe(III) and total Fe, iron was reduced to Fe(II) by adding 2.8 mM ascorbic acid (volumes corresponded to 2% of the sample volume) 10 min prior to analysis.

As demonstrated previously, in the first attempt of µFIA-TLM, which was, however, performed in a stop-flow mode, the formation of Fe(II)(*o*–phen)_3_ complex (ferroin) through complexation reaction of Fe(II) with *o*–phen requires at least 3 min to reach completion in a microfluidic channel [46]. To allow sufficient time for complexation reaction, a second microfluidic chip was added into the microfluidic system shown in Figure 1b, and a 37 cm long capillary (0.8 mm ID) was used as a mixing channel that connected both microchips. The microreactor chip was connected to syringe pumps, while the second microchip served as a platform for TLM detection of ferroin produced after the injection of samples into the carrier stream.

Figure 5 depicts the recorded TLM signal for consecutive injections of samples with Fe(II) concentrations in the 2–20 µM range. The results demonstrate good reproducibility of signals for consecutive injections, which were reflected in within-day relative standard deviations ≤3.9% and ≤8.4% for total Fe and Fe(II), respectively, while between−day reproducibility in terms of RSD was ≤6.4% and ≤10.4% for total Fe and Fe(II), respectively. It is also seen in Figure 5 that the blank signals are about one order of magnitude lower as compared to signals from 2 μM Fe(II), which along with a relatively low standard deviation of blank measurements, provides low limits of detection, as explained in continuation.

Calibration lines were constructed in the 0.1–100 μM concentration range for both Fe(II) and Fe(III). A calibration curve constructed from Fe(III) standard solutions was also used for the determination of total Fe. The calibration lines have shown linearity on the 0.1–70 μM concentration range with correlation coefficients *R*^2^ = 0.9996 for Fe(II) and *R*^2^ = 0.9982 for Fe(III) and total Fe. Limits of detection (LOD), calculated on the S/N = 3 basis, were determined from the standard deviation of the blank signal and the slopes of calibration lines and were found to be 100 nM for Fe(II) and 70 nM total Fe or Fe(III).

Finally, the accuracy of the μFIA-TLM technique was tested by determining the Fe(II), and total Fe content in synthetic marine cloudwater spiked with different amounts of Fe(II) and Fe(III). The results for three replicate injections at each concentration level are presented in Table 2.

The determined concentrations of total Fe, in general, match well the concentrations of added iron, and the accuracy for determination of total iron in terms of analytical recovery was well within the 80–120% range, as recommended for the analytical methods used in the determination of trace metals in waters [47]. However, the accuracy for the determination of Fe(II) in the presence of added Fe(III) is acceptable only at the highest spiking level.

Since the concentrations determined for total Fe show good agreement with the added amounts of total iron, we conclude that recoveries for Fe(II) exceeding 100% are not due to the inaccuracy of the method but arise most probably from sample matrix influenced changes in oxidation states of iron (reduction of Fe(III) to Fe(II) in solutions of synthetic cloudwater containing oxalate), which contributes up to 0.8 µM Fe(II) and, therefore, to higher chemical yield for Fe(II).

The achieved LODs are comparable to those previously reported for stop flow µFIA-TLM, where a detection limit of 2 zeptomoles of Fe(II) (corresponding to 100 nM Fe(II)) was achieved in the detection volume on the microchip [46]. Considering the difference in the optical interaction length, the LODs also compare favorably to those offered by UV−Vis spectrophotometry. For illustration, when using the UV−Vis technique, the LODs in this work were decreased to 10 nM for Fe(II) and total Fe. However, to achieve such improvement, a 1000 times longer optical path than in µFIA-TLM (100 µm) was needed, which necessitates sample volumes of over 25 mL.

The LODs provided by µFIA-TLM are sufficiently low for analysis of cloudwater, which is known to contain iron species at levels higher than 0.1 µM [43,44]. Considering similar limits of detection and very low sample volume requirement of µFIA-TLM (3 µL), this should be the method of choice for determination of Fe(II) and total Fe in investigations of processes in cloudwater, where multiparameter analysis is desired (determination of other ions, ligands, microbial counts, etc.) and large volume cloudwater samples (several 100 mL) cannot be collected.

### 3.3. Determination of Ammonium in Water by µFIA-TLM

Ammonium is another relevant pollutant of global importance, which is also found in cloudwater and, consequently, in rain, contributing to the acidification of soil as well as to increased nitrogen levels in surface waters and forest ecosystems [48]. The concentrations of NH_4_^+^ in cloudwater are in the 100 µM range, while the concentrations found in the natural waters can be as low as a few µM, with 0.5 mg/L (27.8 µM) being the regulatory limit for drinking water and 0.06 mg/L (3.3 µM) for the groundwaters [49]. These levels are already quite challenging for interference-free analytical methods, such as the indophenol blue method, which offers an LOD of 0.16 mg/L (8.9 µM) [50]. For this reason, TLS has already been exploited to improve the LODs of the indophenol method for the determination of ammonia which was reduced down to 0.4 µM [18]. Since the TLS method still required a considerable amount of manual work in the preparation of solutions and a relatively long incubation time (10 min) for the formation of indophenol blue, we aimed to adapt the indophenol method for on-line detection in a microchip by the TLM technique.

Given the differences in spectral characteristics of ferroin (absorption maximum at 520 nm) and indophenol blue (absorption maximum at 625 nm), the TLM microscope presented in Figure 1 was modified by replacing the pump and probe lasers with two solid-state lasers, which provided adequate wavelengths for the given application. A 100 mW (660 nm) CUBE laser (Coherent Inc., Santa Clara, CA, USA) was used as a pump beam source, while an 8 mW (532 nm) laser (Coherent Inc., Santa Clara, CA, USA) provided the probe beam. The dichroic mirror was also replaced to enable transmission at 660 nm and reflection at 532 nm. Before reaching the photodiode, the pump beam was filtered out by a 532 nm interference filter. The given experimental setup offered the highest S/N ratio at 503 kHz modulation frequency.

The carrier solution consisted of commercial bleach (42 g/L of active Cl) and 0.5 M NaOH in the 1:1 volume ratio, with 5% EtOH added to improve the sensitivity (by 20%) and to decrease the baseline noise, as determined experimentally. Prior to injection (5 μL at 50 μL/min injection flow rate) into the carrier stream, the samples or NH_4_^+^ standard solutions were mixed in a 1:1 ratio with a reagent containing 1.5 M sodium salicylate, 30 g/L potassium sodium tartrate, and 2.5 mM MnSO_4_ in the 5:5:1 volume ratio.

As known from previous work on TLS detection of ammonium, the indophenol reaction takes almost two hours to complete [18]. Therefore, optimization of the carrier flow rate was performed to enable good sensitivity and reasonable sample throughput.

As shown in Figure 6a, the signal increases as the carrier flow rate decreases since more time is allowed for the completion of the indophenol reaction. A 90 min reaction time is enabled at 0.2 μL/min. (Figure 6b), which also provides the highest injection peak. The peak, however, extends over 2000 s elution time at the baseline. This is far from acceptable in terms of analysis time as well as sample throughput. However, as evident from the plot in Figure 6a, when increasing the carrier flow rate from 5 μL/min to 10 μL/min, the sensitivity is reduced by a factor of two, while a sample throughput is only increased from 24 to 30 samples/hour, as can be deduced from the widths of the signal peaks at the baseline. Therefore, a carrier flow rate of 5 μL/min was chosen as optimal.

Under the conditions described above (5 μL/min carrier flow rate), the relative standard deviation of TLM signals from five replicate injections of standard solutions (4.5 mM NH_4_Cl) was 8.9%. The calibration with NH_4_Cl solutions in the 0–0.55 mM concentration range revealed the linear range of the µFIA–TLM indophenol method between 0–0.50 mM NH_4_^+^ (*R*^2^ = 0.9977), with LOD (S/N = 3) of 2.3 µM (41 μg/L). This value compares favorably with the LODs achieved previously by the TLS technique [18], considering the 100 times shorter optical length in the case of µFIA-TLM. At the same time, µFIA-TLM offers automation regarding the mixing of the sample with reagents and transport to the detector, which very much improves the reproducibility of the incubation time and of the entire analytical process.

### 3.4. Determination of Iron Species in Water and Polar Sea Ice by DGT-BDS

The determination of iron species in natural waters, sediments, and ice is based on coupling BDS to diffusive gradients in thin-films (DGT) technique which is a sampling technique capable of in situ samplings of the labile fraction of metals in an aqueous medium [51]. Such a sampler consists of a filter layer, an agarose polyacrylamide (APA) diffusive gel, and a resin gel (binding gel), which selectively binds metals and their species (free ions, labile complexes). Such a passive sampler is assembled as described previously for the determination of iron species in river sediments [26] and deployed for a fixed period of time in waters and sediments or simply laid on top of the cut ice core to accumulate ions in its binding gel.

For the purpose of calibration and method validation, binding gels were immersed into solutions containing different concentrations of Fe(II), i.e., from 0 to 1.0 μM for one to five days, and then immersed into 3 mM 1,10 phenanthroline (*o*–phen) solution for five days to enable complete complexation of Fe(II) with *o*–phen. For the determination of Fe(III) and total iron, binding gels were treated with 5.1 mM ascorbic acid to reduce Fe(III) into Fe(II) before complexation with *o*–phen. Next, the gels were transferred onto a glass support and dried for one day at room temperature.

In this work, two types of gels were used to adsorb iron ions from the solution, i.e., a Chelex–100 resin and suspended particulate reagent-iminodiacetate (SPR-IDA). Both of them are semi-transparent gels, which chemically bind the iron ions. SPR-IDA is a polystyrene divinylbenzene substrate chemically derived with iminodiacetate functional groups, which has the same chemical structure as Chelex–100 but has a much smaller initial bead size (0.2 μm) over Chelex–100 (around 100 μm).

The BDS measurements on DGT gels were performed by the use of the experimental setup presented in Figure 2.

#### 3.4.1. Performance Characteristics

To test the performance of the method, the calibration lines for Fe(II) and total Fe were constructed by immersing 1 × 1.5 cm^2^ pieces of Chelex-100 gels into solutions of iron ions in the 0–1.0 μM concentration range. Three pieces of gels were prepared for each concentration, and BDS signals were acquired for each gel at six different points on the surface of the gel by averaging the BDS signal over 1 min of measurement time. The average BDS signal was calculated for signals from all three gels. The calibration lines for both Fe(II) and total Fe determination show a linear relationship over the concentration range of 0–1.0 μM with the correlation coefficient *R*^2^ ≥ 0.995. A typical calibration line is presented in Figure 7.

The relative standard deviation for measurements on gels exposed to 800 nM solution of Fe(II) was 6% which confirms good reproducibility in the preparation and treatment of gels and homogeneous distribution of the analyte over the entire area of the gel. As expected, the homogeneity of Fe distribution in the gel and between the gels is not as good in the lower concentration range. This is also reflected in higher RSD, which was 16% for gels exposed to solutions of 200 nM Fe(II), close to the limit of quantification of the method, i.e., 165 nM.

The calculated LODs (S/N = 3 bases) for Fe(II) determination, expressed as mass of Fe(II), corresponds to 70 ng of Fe(II) adsorbed on the entire DGT gel or approximately 3.3 ng Fe in the volume of the gel under the BDS excitation laser spot (1.5 mm radius) on the gel surface.

These values indicate that DGT-BDS provides low enough LOD for sensing iron in environmental samples such as natural freshwaters, where concentrations of iron are often below 1 μM [52]. The lower concentration range is, however, limited by the LOD of the technique (20 nM), which under conditions applied in this work (five-day deployment of the sampler), does not allow for sensing of ocean waters, where the concentrations could be as low as 0.2 nM [53]. It must, however, be underlined that the lowest concentration detectable by DGT-BDS depends on the lowest detectable mas of Fe adsorbed on the entire gel (70 ng in our case), but also on the time interval of the deployment of the sampler, as shown by Equation (3) [54].
(3)$$C_{\min}=\frac{M_{\min}\Delta d}{{D\;A}_{\mathrm{es}}\;t}$$
where C_min_ is the minimum detectable concentration (LOD), M_min_ is the minimum detectable mass of adsorbed analyte on the gel with a surface A_es_ (1.5 × 10^−4^ m^2^ for the gels used for calibration in this work), D is the diffusion coefficient (5.9 × 10^−10^ m^2^ s^−1^ for labile Fe species in Chelex−100), ∆d is the diffusive layer thickness (0.67 mm for Chelex−100 gels), and t is the time interval of sampler deployment.

Deployment of a DGT passive sampler for several months would at least theoretically enable the detection of the lowest concentrations of Fe, which can be encountered in ocean waters (below 1 nM) [53].

As the final step in the validation of the DGT-BDS method, the analytical yields were calculated for the determination of Fe(II) in 200 nM–1.00 μM solutions. The results are presented in Table 3.

The results show that the analytical yield of the method is satisfactory (93–99%) and in agreement with US−EPA recommendations for acceptable analytical yields for the determination of metals and trace elements in water (80–120%) [47].

#### 3.4.2. Determination of Fe(II) and Total Fe Distribution in Polar Sea Ice

BDS was also used to determine the concentration of iron on the passive sampler gel exposed to the sea ice samples from the Davis Station, situated on the coast of Cooperation Sea in Princess Elizabeth Land, Ingrid Christensen Coast in the Australian Antarctic Territory.

SPR-IDA resin gels (2 × 3 cm^2^) were used in this case (D = 2.96 × 10^−10^ m^2^ s^−1^, ∆d = 0.05 mm) and were deployed on sea ice samples for 24 h in Glaciology Laboratory at Universite Libre de Bruxelles at −4 °C. Gels were washed with double-deionized H_2_O after the deployment, stored in plastic zip-lock bags, and transported to the laboratory of the University of Nova Gorica for further analysis by BDS, as described in the previous section. Each gel was cut into four smaller pieces (approximately 1 cm by 1.5 cm each) to fit into the BDS spectrometer. Each piece of binding gel was treated as described above in Section 3.4 and in our previous work [26]. The concentration of Fe(II) or total Fe was determined by performing measurements at nine measurement points on each piece of gel, with an about 3 mm lateral resolution, scanning the entire surface of the gel actually.

It shall be noted that SPR-IDA gels provided about 1.6 times lower RSD, which is attributed to better homogeneity of Fe distribution within the gel due to the much smaller size of gel beads of SPR-IDA, as compared to Chelex–100. As a result, an LOD of 30 ng of Fe on 1 cm^2^ of exposed gel was achieved. Using equation 1, this can be converted into an average concentration of 10 nM Fe in water. The determined values for iron in the sea ice sample, expressed in terms of concentrations in melted ice in contact with DGT gel, were in the range of <LOD − 280 nM and <LOD − 250 nM for Fe(II) and total Fe, respectively. A graphical representation of Fe(II), as well as total Fe concentration distribution over the surface of SPR-IDA resins after their deployment on sea ice surface, is given in Figure 8.

## 4. Conclusions

It has been demonstrated by applications presented in this paper that photothermal techniques, TLM and BDS in particular, provide highly sensitive and specific tools for the environmental sensing of liquid and solid samples. This includes relatively exotic examples of samples that are not easy to collect, such as cloudwater (the liquid fraction of clouds) and polar sea ice. At the same time, TLM, in combination with microfluidics, offers low sample volume capability as well as fast analytical response and high sample throughput, such as required for fast screening systems. Little need for sample handling, as offered by the FIA, is advantageous when determining chemically unstable analytes, which do not allow for sample treatment prior to the analysis. Similar advantages are offered by the BDS technique due to its non-contact capability, which was successfully utilized for the analysis of gels on GDT passive samplers. Most of the presented work focused on the determination of Fe and its redox species (Fe(II), Fe(III)) by μFIA-TLM and DGT-BDS, while determinations of microcystin-LR and ammonium have additionally confirmed the broad applicability of μFIA-TLM for environmental sensing of waters.

It has been demonstrated that using only 1–5 μL of sample μFIA-TLM provides LODs at the level of 100 pg/mL for analytes that undergo fast colorimetric reactions, such as in the case of microcystin-LR in water (LOD = 80 pg/mL, 80 pM). LODs for analytes that require longer times for completion of colorimetric reactions are expectedly higher, like 6 ng/mL for Fe(II) (100 nM) and 41 ng/mL for ammonium ion (2.3 μM). LODs in such cases can be improved by sacrificing the sample throughput, which, however, even at 30 samples per hour, still remains far from four samples/min achieved for triplicate sample injections for the determination of microcystin-LR.

While being known as a less sensitive technique for analysis of liquid samples compared to TLM, BDS, in combination with DGT passive samplers, provided a platform for the determination of iron redox species (Fe(II), Fe(III)) and total Fe in water samples at concentrations one order of magnitude lower as compared to μFIA-TLM. This is attributed primarily to the preconcentration of Fe species on the passive sampler, which, however, requires the deployment of the sampler in water for a period of 24 h or several days. It should not be disregarded that such speciation studies of chemically sensitive systems are not possible by standard analytical techniques such as ICP−OES and ICP−MS, which require aggressive/destructive treatment of the sample prior to analysis and are, therefore, suitable only for determination of total analyte concentration. On the other hand, the non-contact capability of BDS offers an analysis of DGT gels after simple treatment with a complexing agent (*o*−phen), which at the same time stabilizes Fe(II) in the sample. A practical illustration of the capabilities of DGT-BDS is the mapping of Fe(II) and total Fe on a sea ice core. The presented results clearly confirmed differences in the distribution of Fe, extending over two orders of magnitude, while Fe(II) was found to be the dominant Fe species over the core surface.

Given the versatility of μFIA-TLM and DGT-BDS, novel applications of these analytical platforms for ES are expected in the future, taking advantage also of the availability of new specific antibodies, which could enable the detection of pathogens such as viruses in natural waters, while progress in nanotechnology could facilitate the operation of microfluidic systems, by immobilizing analyte−specific molecules as receptors on magnetic nanoparticles, which are easily loaded as well as removed from the microchips after depletion. Such research related to μFIA-TLM is already underway in the authors’ laboratory [55], while possibilities for improvement of the versatility of μFIA-TLM and DGT-BDS were already presented particularly by utilization of incoherent light sources [56,57].

At the same time, efforts are being made to improve the sensitivity of BDS and TLM techniques. These efforts are, on one side, related to the development of novel instruments with incorporated resonant cavities, which enable multiple passing of the probe beam. In such a case, the improvement in sensitivity of TLM is approximatively proportional to the number of passes of the probe beam through the sample [57] or through the TOs above the sample in the case of BDS [38,58]. In the case of TLM, improvements in sensitivity and corresponding LOD of up to 10 times are forecasted by constructing microchips from materials with large dn/dT and low k values. As predicted theoretically and confirmed by the experiment in the case of TLM, the generated thermal field extends from the carrier liquid in the microchannel into the adjacent material of the microchip by a distance comparable to the dimensions of the microchannel. Consequently, additional thermal lenses are generated on the top and bottom sides of the microchannel, which enhances the defocusing of the probe beam [36,56,59].

## Figures and Tables

**Figure 1 sensors-23-00472-f001:**
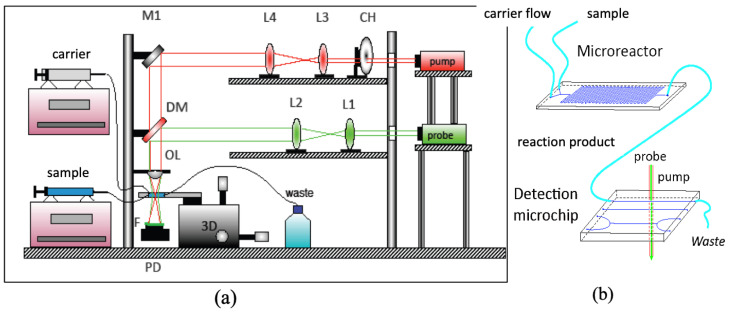
Schematic diagram of a laser-excited TLM (**a**) coupled with a microfluidic system (**b**). CH: mechanical chopper; L1–L4: lenses; M1: mirror; DM: dichroic mirror; OL: objective lens; F: 633 nm interference filter; PD: photodiode; 3D: xyz translation stage.

**Figure 2 sensors-23-00472-f002:**
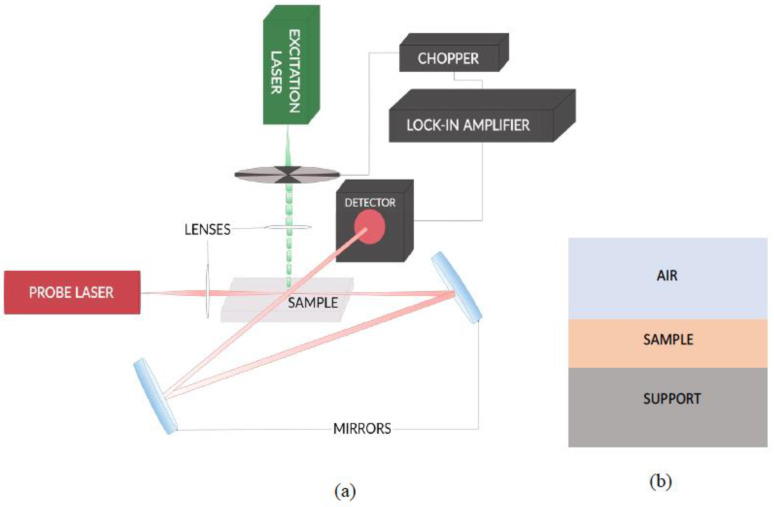
Scheme of the BDS experimental setup used in this work (**a**) and the arrangement of the sample position with respect to the support and the fluid (air) (**b**).

**Figure 3 sensors-23-00472-f003:**
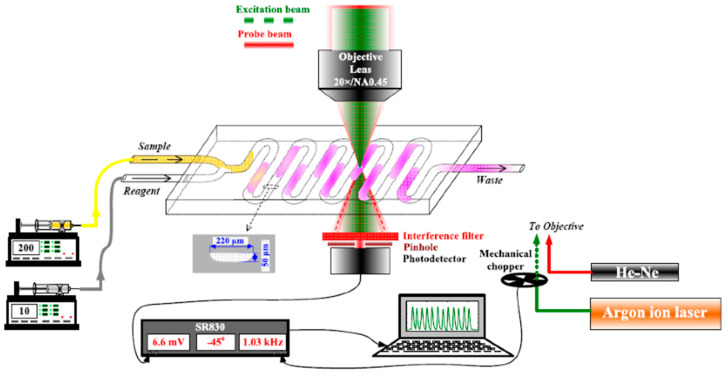
Schematic presentation of a µFIA-TLM detection platform for determination of microcystin-LR.

**Figure 4 sensors-23-00472-f004:**
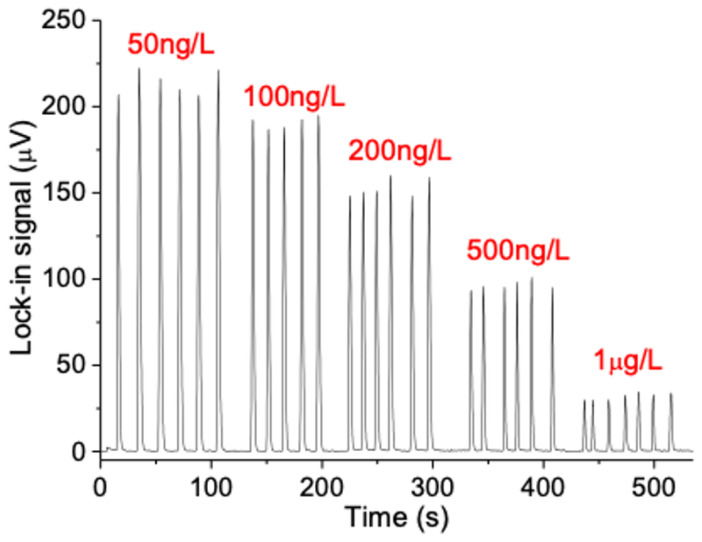
µFIA-TLM signals recorded for replicate injections of microcystin-LR standard solutions (concentrations indicated above the peaks in red).

**Figure 5 sensors-23-00472-f005:**
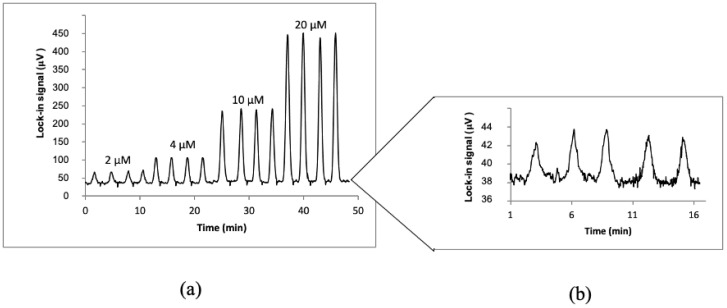
μFIA-TLM signals for four replicate injections of four concentration levels of Fe(II) (sample injection volume: 3 µL, carrier flow rate: 5 µL/min) (**a**). The insert (**b**) on the right shows an expanded view of signals for injections of the blank.

**Figure 6 sensors-23-00472-f006:**
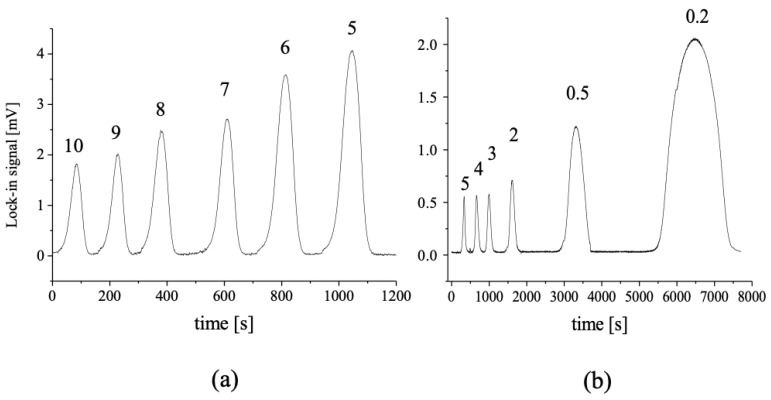
Dependence of the TLM signal on the carrier flow rate (indicated on top of the peaks in μL/min) at 5 μL injection volume from 5 mM NH_4_Cl (**a**) and 0.5 mM NH_4_Cl (**b**). Lower concentration was chosen at lower flow rates for the prevention of signal saturation.

**Figure 7 sensors-23-00472-f007:**
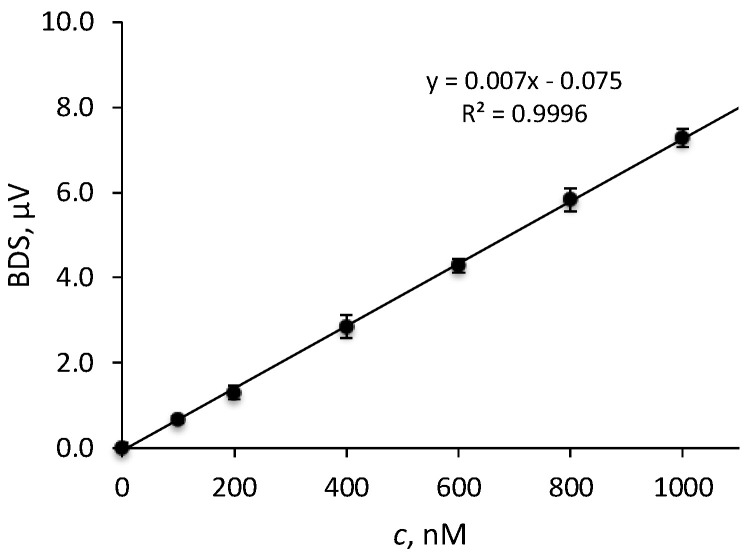
Calibration line for Fe(II) determination by BDS−DGT technique using a Chelex−100 gel.

**Figure 8 sensors-23-00472-f008:**
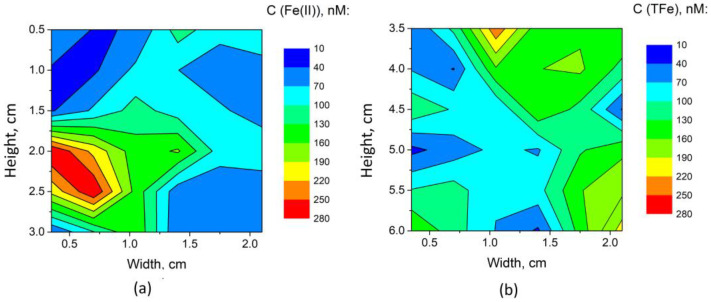
Graphical representation of Fe(II) (**a**) and total Fe (TFe) (**b**) distribution over the surface of SPR-IDA, reflecting the distribution of Fe species on the sea ice core surface. Representation of the concentrations on the sea ice surface was prepared on the basis of 36 measurement points over the area of each gel using the ORIGIN software (Version 9.6.5 OriginLab Corporation, MA, USA) and assuming linear changes of concentrations between the measurement points.

**Table 1 sensors-23-00472-t001:** Comparison of figures of merit for the μFIA-TLM PP2A method and existing methods for detection of microcystin-LR.

Detection Method	LOD	Linear Range	Sample Throughput/ Analysis Time	Ref.
μFIA-TLM PP2A	80 ng/L	0.08–1 μg/L	4 samples/min (triplicate injection)	This work
ELISA	0.1 μg/L	0.15–5 μg/L	150 min/96 samples	[40]
PP2A microtiter plate	0.25 μg/L	0.25–2.5 μg/L	30 min/96 samples	[41]
Immuno−electrode	0.01 ng/L	0.0001–0.1 μg/L	37 min/sample	[42]

**Table 2 sensors-23-00472-t002:** Determination of iron in spiked artificial cloudwater by μFIA-TLM.

Added Fe(II) + Fe(III), μmol/L	Determined Fe(II), μmol/L	Recovery Fe(II),	Determined Total Fe, μmol/L	Recovery Total Fe,
0.5 + 0.5	0.65 ± 0.01	130 ± 1	1.05 ± 0.01	105 ± 1
2 + 2	2.8 ± 0.2	140 ± 8	4.09 ± 0.09	102 ± 2
10 + 10	10.8 ± 0.3	108 ± 3	20.5 ± 0.7	102 ± 3

**Table 3 sensors-23-00472-t003:** The analytical yield of the DGT-BDS method for determination of Fe(II) in water.

Added Concentration, nM	Measured Concentration, nM	Yield, %
200	187 ± 30	93 ± 15
600	577 ± 45	96 ± 8
1000	992 ± 60	99 ± 6

## Data Availability

Not applicable.

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
