# Peer review of "Recent Progress and Applications of Thermal Lens Spectrometry and Photothermal Beam Deflection Techniques in Environmental Sensing"

_sensors, 2023, doi:10.3390/s23010472_

Round 1

Reviewer 1 Report

The work is relevant since it reviews some recent works and new applications of the TLS and BDS techniques in environmental sensing, a very important current topic worldwide. I support the publication of this work, but I have a few questions that could be addressed by the authors before publication.

1) In the introductory section the authors say that TLS and BDS are the best suited among all PT methods for analysis of condensed matter samples. A brief justification for this assertion should be added because it is true that TLS has been emerged as the option of choice among all PT methods, but in particular for ES-applications.  However, as the authors mention, analytical spectroscopic applications of BDS are very scarce.  It is worth mentioning that PAS has been also proposed by some authors for such applications, but to a lesser extent.

2) In another paragraph the authors mention that BDS relies on probing the TOs induced in a fluid close to the sample surface. The same idea can be deduced from the schema shown in Fig. 2. I think this is the configuration widely used mainly for thermal characterization. But, is this really the better configuration for the applications of TLS described here. What was the fluid at the sample’s surface in this case? Air is not a suitable ones because its small dn/dT coefficient. I think that more details about the BD measurements should be added including on samples preparation. Were the Iron species detected as they are or was some colorimetric method used? If not, what about the influence of competing species? I also suggest showing the signal vs Fe concentration curves used for LOD determinations (at least a typical ones).

3) Considering the optical transparency of the studied samples by BDS, couldn´t it be better to use in these cases a configuration in which the probe beam axis is offset from the excitation point inside the sample, as described for example by Couch et al [B Couch et al Methods Appl. Fluoresc.7 (2019) 015004]?

4) Section 3.1. Why was the 3 kHz frequency chosen?

Author Response

1) In the introductory section the authors say that TLS and BDS are the best suited among all PT methods for analysis of condensed matter samples. A brief justification for this assertion should be added because it is true that TLS has been emerged as the option of choice among all PT methods, but in particular for ES-applications.  However, as the authors mention, analytical spectroscopic applications of BDS are very scarce.  It is worth mentioning that PAS has been also proposed by some authors for such applications, but to a lesser extent.

Clarification on the advantage of TLS and BDS for detection in condensed samples is given, along with mention of PAS applications (lines 50-57).

2) In another paragraph the authors mention that BDS relies on probing the TOs induced in a fluid close to the sample surface. The same idea can be deduced from the schema shown in Fig. 2. I think this is the configuration widely used mainly for thermal characterization. But, is this really the better configuration for the applications of TLS described here.

This comment is not clear. The experimental configuration for TLS (TLM in our case) is presented on Fig. 1 while the experimental set up for BDS is indeed shown on Fig. 2. in our original manuscript.

What was the fluid at the sample’s surface in this case? Air is not a suitable ones because its small dn/dT coefficient. I think that more details about the BD measurements should be added including on samples preparation.

The relevant explanations are given at the end of section 2 (Instrumental) - lines196-201, as well as in the context of the answer to next question (see below), including reference to a previous work (ref. 26). The arrangement of the sample position with respect to the support and the fluid (air) is added in Fig. 2b for clarity.

Were the Iron species detected as they are or was some colorimetric method used? If not, what about the influence of competing species?

Fe species were determined based on the colorimetric reaction of Fe(II) with 1,10 phenanthroline, therefore no interferences are expected from competing species. The analytical procedure is now better explained in lines 431-434 and lines 510-511.

 I also suggest showing the signal vs Fe concentration curves used for LOD determinations (at least a typical ones).

To present a typical calibration line for BDS measurements of Fe(II) a new Figure 7 was inserted after line 455

3) Considering the optical transparency of the studied samples by BDS, couldn´t it be better to use in these cases a configuration in which the probe beam axis is offset from the excitation point inside the sample, as described for example by Couch et al [B Couch et al Methods Appl. Fluoresc.7 (2019) 015004]? 

The presented BDS configuration in Ref. (B Couch et al Methods Appl. Fluoresc.7 (2019) 015004) is suitable for transparent samples. In our case the analyte (Fe(II)) is not determined in solution but adsorbed on the resin. The BDS measurements are performed on Chelex-100 and SPR-IDA resins which are semi-transparent and strongly scatter the probe beam. It is this impossible to apply the above mentioned (collinear) configuration of BDS technique.

4) Section 3.1. Why was the 3 kHz frequency chosen?

The explanation is given in lines 220-222.

Reviewer 2 Report

The manuscript is mostly well-written and I recommend publication after the authors address the following points:

1- The Intro could use some more references.

2- In photothermal techniques, the obtained spectra can be difficult to discern for mixtures. The authors are encouraged to discuss what spectral processing can be more effective to separate the peaks.

3- In thermal lensing or laser reflectometry, environmental noise (in liquid or gas) and other noise sources can make low concentration detection challenging. In the case of stochastic noise in the spectra, one can for example use KLT denoting algorithm which is highly suitable for the presented measurement of low concentration. I encourage the authors to cite the work of Zaharov:

Zaharov, VV, et al., “Karhunen-Loeve treatment to remove noise and facilitate data analysis in sensing, spectroscopy and other applications,” ANALYST 139(22), 5927-5935, (2014); DOI10.1039/c4an01300j

4- Can the authors clearly state what a reasonable range for LOD for the employed techniques and how the LOD can be improved.

5- I recommend the authors also include the basic thermal lensing and laser deflection equations (no need for derivation). This would help exposing the physics for the benefit of the readers.

Author Response

1- The Intro could use some more references.

Additional references were added and some outdated references were replaced by more recent ones. They can be found marked in red in a copy of the manuscript where all the corrections are highlighted. Some references do however refer to basic texts in the field and cannot be replaced.

2- In photothermal techniques, the obtained spectra can be difficult to discern for mixtures. The authors are encouraged to discuss what spectral processing can be more effective to separate the peaks.

We performed all measurements at fixed wavelengths offered by the lasers we used, and we did not record any absorption spectra. Spectral processing was therefore not needed in our applications, while eventual possible interferences were eliminated by using specific and selective colorimetric reactions and reagents (1,10 phenanthroline for Fe(II), Indophenol blue for ammonium and PP2A inhibition for microcystin)

3- In thermal lensing or laser reflectometry, environmental noise (in liquid or gas) and other noise sources can make low concentration detection challenging. In the case of stochastic noise in the spectra, one can for example use KLT denoting algorithm which is highly suitable for the presented measurement of low concentration. I encourage the authors to cite the work of Zaharov: Zaharov, VV, et al., “Karhunen-Loeve treatment to remove noise and facilitate data analysis in sensing, spectroscopy and other applications,” ANALYST 139(22), 5927-5935, (2014); DOI10.1039/c4an01300j

As argued above, our applications do not require smoothing of the photothermal spectra since we do not deal with them. We therefore see no need to cite the suggested reference.

4- Can the authors clearly state what a reasonable range for LOD for the employed techniques and how the LOD can be improved.

To our opinion LODs are already clearly presented in the original manuscript for each of the analytes and applications, and are additionally summarized in conclusions, along with indications of possible improvements (now lines 544 -551). Still, we have additionally underlined the options for improvements of sensitivity in the future outlook, which we gave at the end of Conclusions and is now expanded by lines 577-589.

5- I recommend the authors also include the basic thermal lensing and laser deflection equations (no need for derivation). This would help exposing the physics for the benefit of the readers.

Equations for the signals in case of TLS and BDS presented and explained in lines 85-110.

Round 2

Reviewer 1 Report

The authors have considered my comments in preparing this new version of the manuscript and I believe it has been sufficiently improved to warrant publication in Sensors.